# Frontal cortex hyperactivation and gamma desynchrony in Fragile X syndrome: Correlates of auditory hypersensitivity

Ernest V. Pedapati[1,2,3☯*], Lauren E. Ethridge[4,5☯], Yanchen Liu[1], Rui Liu[1], John A. Sweeney[3], Lisa A. DeStefano[6], Makoto Miyakoshi[1], Khaleel Razak[7], Lauren M. Schmitt[6,8], David R. Moore[9,10], Donald L. Gilbert[2,8], Steve W. Wu[2,8], Elizabeth Smith[6,8], Rebecca C. Shaffer[6,8], Kelli C. Dominick[1,3], Paul S. Horn[2], Devin Binder[11], Craig A. Erickson[1,3]

1 Division of Child and Adolescent Psychiatry, Cincinnati Children's Hospital Medical Center, Cincinnati, Ohio, United States of America, 2 Division of Neurology, Cincinnati Children's Hospital Medical Center, Cincinnati, Ohio, United States of America, 3 Department of Psychiatry, University of Cincinnati College of Medicine, Cincinnati, Ohio, United States of America, 4 Department of Pediatrics, Section on Developmental and Behavioral Pediatrics, University of Oklahoma Health Sciences Center, Oklahoma City, Oklahoma, United States of America, 5 Department of Psychology, University of Oklahoma, Norman, Oklahoma, United States of America, 6 Division of Developmental and Behavioral Pediatrics, Cincinnati Children's Hospital Medical Center, Cincinnati, Ohio, United States of America, 7 Department of Psychology, University of California, Riverside, California, United States of America 8 Department of Pediatrics, University of Cincinnati College of Medicine, Cincinnati, Ohio, United States of America, 9 Communication Sciences Research Center, Cincinnati Children's Hospital Medical Center, Cincinnati, Ohio, United States of America, 10 Division of Biomedical Sciences, School of Medicine, University of California, Riverside, California, United States of America, 11 Manchester Centre for Audiology and Deafness, University of Manchester, Manchester, United Kingdom,

☯ These authors contributed equally to this work.
* ernest.pedapati@cchmc.org

## Abstract

Fragile X syndrome (FXS) is an X-linked disorder that often leads to intellectual disability, anxiety, and sensory hypersensitivity. While sound sensitivity (hyperacusis) is a distressing symptom in FXS, its neural basis is not well understood. It is postulated that hyperacusis may stem from temporal lobe hyperexcitability or dysregulation in top-down modulation. Studying the neural mechanisms underlying sound sensitivity in FXS using scalp electroencephalography (EEG) is challenging because the temporal and frontal regions have overlapping neural projections that are difficult to differentiate. To overcome this challenge, we conducted EEG source analysis on a group of 36 individuals with FXS and 39 matched healthy controls. Our goal was to characterize the spatial and temporal properties of the response to an auditory chirp stimulus. Our results showed that males with FXS exhibit excessive activation in the frontal cortex in response to the stimulus onset, which may reflect changes in top-down modulation of auditory processing. Additionally, during the chirp stimulus, individuals with FXS demonstrated a reduction in typical gamma phase synchrony, along with an increase in asynchronous gamma power, across multiple regions,

**Data availability statement:** All deidentified data underlying the findings reported in this manuscript are publicly available in the following repositories: - Raw EEG data from the study "Auditory EEG Biomarkers in Fragile X Syndrome: Clinical Relevance" (#1227) are stored in the National Database for Autism Research (NDAR), while source-localized tracings in EEGLAB SET format are deposited in Zenodo and can be accessed at 10.5281/zenodo.14967041. - Custom code and scripts used to analyze data are available in a public GitHub repository at https://github.com/cincibrainlab/vhtp. - Figures and analysis data tables are deposited in Figshare and can be accessed at 10.6084/m9.figshare.19435496. These datasets and materials are provided in standard formats with accompanying documentation to facilitate reuse and replication. All data is deidentified and unrestricted, in accordance with PLOS ONE's open data policy and our institutional ethics guidelines.

**Funding:** The present study was federally funded by the National Institutes of Health (NIH) Fragile X Center (U54HD082008, PI: John Sweeney; U54HD104461, PI: Craig Erickson). The funders had no role in study design, data collection and analysis, decision to publish, or preparation of the manuscript

**Competing interests:** No authors have competing interests.

most strongly in the temporal cortex. Consistent with these findings, we observed a decrease in the signal-to-noise ratio, estimated by the ratio of synchronous to asynchronous gamma activity, in individuals with FXS. Furthermore, this ratio was highly correlated with performance in an auditory attention task. Compared to controls, males with FXS demonstrated elevated bidirectional frontotemporal information flow at chirp onset. The evidence indicates that both temporal lobe hyperexcitability and disruptions in top-down regulation play a role in auditory sensitivity disturbances in FXS. These findings have the potential to guide the development of therapeutic targets and back-translation strategies.

## Introduction

Fragile X Syndrome (FXS) is an X-linked neurodevelopmental disorder caused by methylation-based silencing of fragile X messenger ribonucleoprotein (FMRP) [1]. Individuals with FXS typically experience intellectual disability, anxiety, and sensory hypersensitivity, specifically hyperacusis, which is an increased sensitivity to sounds. Auditory hypersensitivity can have a significant impact on daily functioning and may present as difficulty tolerating certain sounds, heightened sensitivity to loud noises, and an overall increased sensitivity to auditory stimuli [2]. Studies using electroencephalography (EEG) have provided insight into the temporal abnormalities of auditory processing in FXS [3–5]. However, it remains poorly understood how different regions of the brain contribute to hyperacusis in FXS [6].

Interrogation of the auditory system using the chirp stimulus (broadband noise amplitude modulated by a sinusoid linearly increasing in frequency from 0–100 Hz over 2 seconds) is a valuable method to examine neural synchronization and background response across a range of frequencies, typically leading to phase alignment within the gamma frequency range (30–80 Hz) [7]. This precise phase alignment of gamma oscillations contributes to sensory and higher-order processes [8,9]. Gamma oscillations that are locked to stimuli play an important role in filtering and amplifying the processing of sensory stimuli [10,11]. Alternatively, we hypothesized that increases in asynchronous gamma activity may represent "noise" that disrupts the auditory cortex's ability to contextually modulate and precisely align gamma oscillations to task demands.

Auditory evoked potential (AEP) studies in FXS have reproducibly demonstrated impairments in gamma phase synchronization and elevations in background gamma power when using the chirp stimulus [12–14]. These alterations were correlated with cognitive measures including auditory attention, processing speed, and cognitive flexibility [14]. Indeed, in FXS, in addition to gamma phase synchrony deficits, asynchronous elevations in gamma power have been observed at rest [15–18], during speech tasks [19], and during auditory tasks [12,14]. These similar alterations in gamma synchronization and asynchronous gamma power can be observed in $Fmr1^{-/-}$ KO mouse which is the primary animal model for FXS [20,21]. Given that FMRP regulates synapses in an activity- and stimulus-dependent manner, it can be hypothesized that

the absence or reduction of FMRP impairs the precision and efficiency of neural firing, particularly in regions and networks that experience high cognitive or sensory processing demands [22,23].

Recent studies, including those conducted in ASD, have found that conditioned top-down activity originating in the frontal cortex provides contextual cues to modulate auditory system sensitivity [24–28]. However, the specific contribution of different brain regions to auditory hypersensitivity in FXS remains unclear. This is primarily due to limitations of EEG analysis at the scalp level, which may not accurately reflect auditory cortex activity due to its tangential overlap with frontal sources [29]. By examining the spatial distribution of gamma abnormalities during the auditory chirp, we can enhance our understanding of the anatomical origins of sensory dysfunction in FXS and establish systems-level correlates with phenotype [3–5,30].

In this study, we examined source-localized EEG responses to the auditory chirp to characterize spatial and time-domain characteristics of primarily gamma band synchronization and background power. We conducted the study in a well-powered sample of individuals with FXS as well as sex- and age-matched typically developing controls. Our hypothesis was that individuals with FXS would exhibit chirp-associated phase locking deficits, primarily localized to the temporal regions. This was based on the understanding that frequency-following response typically reflects basic bottom-up sensory processing from subcortical and sensory cortical sources [31]. Additionally, we hypothesized that the onset of the chirp stimulus, which engages contextual cueing and salience processing, would elicit a strong response in the frontal cortex [32]. As thalamocortical dysrhythmia is implicated in Fragile X syndrome, likely contributing to atypical neural oscillations, we examine alpha oscillations as possible markers of such deeper disruptions in thalamocortical circuitry and known neuroimaging abnormalities across the cingulate cortex, amygdala, inferior colliculus, and other subcortical structures [33 ]. Furthermore, we expected to observe sex differences in individuals with FXS, consistent with varying levels of FMRP expression [33]. Finally, we predicted that these neurophysiological alterations would be associated with neurocognitive and behavioral correlates, particularly impairments in an auditory attention task that incorporates both top-down and sensory aspects.

## Materials and methods

### Participants

A methodological overview of the study procedures is depicted in Fig 1. Thirty-six participants with FXS (23 males; age, 10–53 years; mean age, 25.4 ± 10.3 years) and 39 age- and sex-matched typically developing controls (22 males; age, 12–57 years; mean age, 28.0 ± 12.2 years) participated in this study (see Table 1). The age and proportion of each sex did not differ between groups. Scalp-level electrode analyses of this dataset were previously published [14]. In our study, all participants gave written informed consent. For minors aged 11–17, both the child/teen provided assent and a parent or guardian provided written consent. No waivers for parental consent were granted by the IRB. All procedures were approved by the Cincinnati Children's Hospital Institutional Board (IRB 2015–8425). Recruitment for the study occurred under NIH FXS Center (U54HD082008) grant began on 1/18/2016 and completed 5/31/2020. Typically developing control (TDC) participants were screened for prior diagnosis or treatment for neuropsychiatric illness, hearing loss, psychiatric medication use, or first-degree relatives affected by serious psychiatric or neurodevelopmental disorders. Participants with FXS were excluded if they had a history of hearing impairment, epilepsy, or recent administration of anticonvulsant medications (including benzodiazepines) that could confound EEG interpretation. To ensure a representative sample of the FXS population [34], participants on stable doses (at least four weeks) of psychiatric medicines were permitted.

### Psychological measures

Participants' cognitive abilities (IQ) were estimated using the Stanford-Binet Intelligence Scales 5th Ed. Abbreviated IQ test. To account for floor effects, deviation scores were calculated for verbal and non-verbal IQ in lower ranges [35].

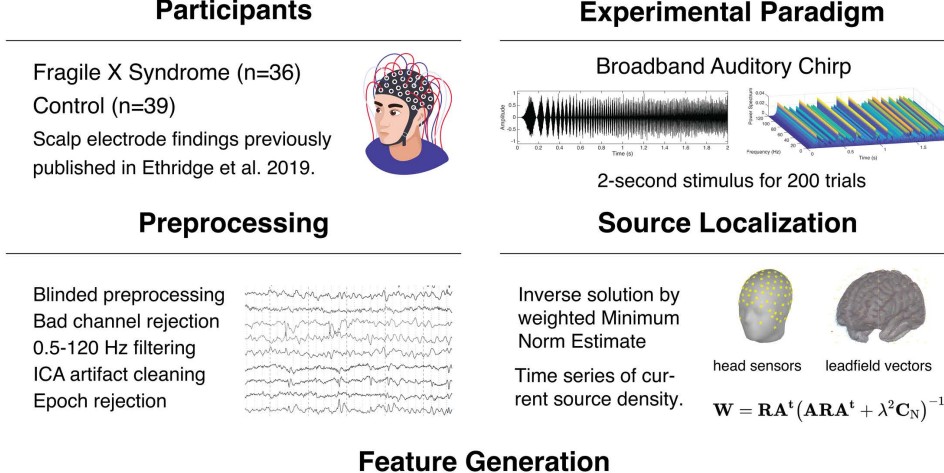

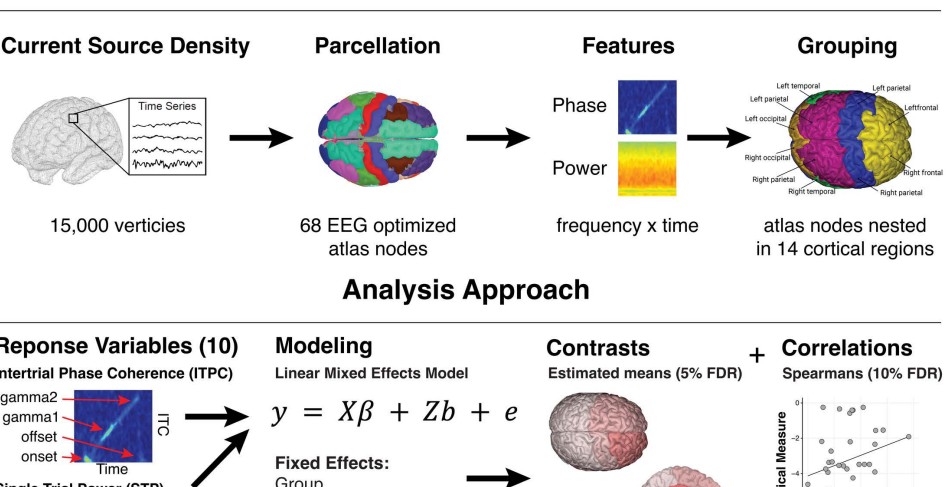

**Fig 1. Overview of the principal methodology.** An auditory evoked potential case-control study using dense array EEG, source localization, and advanced statistical modeling to identify spatiotemporal neural responses to the chirp stimuli and clinical correlations.

Selective speech-sound discrimination was measured in participants using the Woodcock-Johnson III Tests of Cognitive Abilities Auditory Attention (WJ3) [36]. Caregiver assessments of FXS participants included the Social and Communication Questionnaire (SCQ), Anxiety Depression and Mood Scale (ADAMS) [37], and Aberrant Behavior Checklist-Community (ABC-C, optimized for FXS) [38].

## Experimental stimulus

Auditory stimuli were presented diotically through Sony MDR-V150 headphones (65 dB SPL RMS) using Presentation 22 (Neurobehavioral Systems) in a sound-attenuated room. Participants viewed a silent video during the task to facilitate protocol compliance and promote quiet wakefulness [13,15].

**Table 1. Demographics and clinical features of study participants by group.**

| Measure | Group | | p-value[2] |
|---|---|---|---|
| | FXS, N = 36[1] | TDC, N = 39[1] | |
| Sex | | | .5 |
| F | 13 (36%) | 17 (44%) | |
| M | 23 (64%) | 22 (56%) | |
| Age at Visit | 25.4 ± 10.3 | 28.0 ± 12.2 | .5 |
| Deviation IQ | 42.4 ± 29.1 | 102.9 ± 8.3 | <.001 |
| NVIQ | 31.9 ± 36.1 | 103.6 ± 11.2 | <.001 |
| VIQ | 54.9 ± 28.3 | 102.3 ± 1.8 | <.001 |
| ADAMS General Anxiety | 7.7 ± 4.8 | 2.4 ± 2.6 | <.001 |
| ADAMS Obsessive/Compliance Behavior | 2.4 ± 2.3 | .5 ± 1.3 | <.001 |
| SCQ Total | 14.0 ± 7.9 | 2.2 ± 2.4 | <.001 |
| ABC FXS subscale 1: Irritability/aggression | 11.2 ± 11.4 | .6 ± 2.4 | <.001 |
| ABC FXS subscale 4: Hyperactivity/Noncompliance | 7.7 ± 5.9 | .9 ± 2.1 | <.001 |
| ABC FXS subscale 5: Inappropriate speech | 4.4 ± 3.5 | .2 ± .6 | <.001 |
| ABC FXS subscale 2: lethargy/social withdrawal | 5.5 ± 4.4 | .6 ± 1.5 | <.001 |
| ABC FXS subscale 3: stereotypy | 4.0 ± 4.5 | .1 ± .4 | <.001 |
| WJ3 | 65.2 ± 17.9 | 91.9 ± 11.7 | <.001 |

Abbreviations: FXS, Fragile X Syndrome; TDC, typically developing control; F, female; M, male; NVIQ, Non-verbal intelligence quotient; VIQ, verbal intelligence quotient; SCQ, Social Communication Questionnaire; ABC, Aberrant Behavior Checklist; WJ-3, Woodcock III Tests of Cognitive Abilities; ADAMS, Anxiety, Depression, and Mood Scale.

1 Values are presented as n (%) or Mean±SD.

2 Pearson's Chi-squared test; Wilcoxon rank-sum test.

An auditory chirp stimulus was employed to evaluate the integrity of functional cortical responses [7]. The stimulus consisted of a pink noise carrier that is amplitude modulated by a sinusoid linear chirp which increased in frequency from 0 to 100 Hz over 2000 ms (S1 Fig). Participants were presented with 200 repetitions, with each repetition having an interstimulus interval randomly varied between 1500 and 2000 milliseconds. Cortical responses to the chirp were measured using intertrial phase coherence (ITPC), which indicates the strength and consistency of neural phase locking to the stimulus. Control participants typically demonstrate a strong attunement to the current frequency of the chirp, as evidenced by high ITPC values across the frequency range of the chirp [7].

## EEG acquisition and preprocessing

EEG was continuously recorded and referenced to Cz with a sampling rate of 1000 Hz using a NetAmp 400 (Magstim/EGI, Eugene, Oregon) with a Hydrocel 128-channel geodesic saline-based electrode net. Raw data were deidentified and visually inspected offline. Data were digitally filtered from 0.5–120 Hz (12 and 24 dB/octave roll-off, respectively; zero-phase; 60 Hz notch). Bad sensors were interpolated (up to 5% per subject, no more than two adjacent sensors) using spherical spline interpolation implemented in BESA 6.1 (MEGIS Software, Grafelfing, Germany). Ocular, cardiac, and muscle movement artifacts were removed blind to participant group status using independent components analysis (extended infomax) implemented in EEGLAB [39] using MATLAB 2021a (MathWorks, Natick, MA). Segments of data with large movement artifacts were removed before ICA to facilitate algorithm convergence. Data were then transformed to average reference and segmented into 3250 ms trials (-500 ms to 2750 ms). Any trial with post-ICA amplitude exceeding 120 µV was considered as containing residual artifact and removed.

## EEG analyses

**Source localization.** A depth-weighted minimum norm estimate (MNE) model in Brainstorm was used to estimate a distributed source model [40]. Several methods have been developed to calculate dipolar magnitudes from scalp potentials and electrode positions [41]. MNE identifies a solution that minimizes the influence of total brain power and corresponds to Tikhonov regularization [42]. To account for the bias of weaker surface sources towards electrodes in the classical MNE solution, a depth weighting (order,.5; maximal amount, 10) was applied to normalize source amplitude values [43]. The model was constrained to dipoles normal to the cortical surface to optimize for dense array EEG analysis, and an identity matrix was used for noise covariance. Scalp electrodes were co-registered to the ICBM152 common brain template [44]. The forward model, or lead field matrix, was computed using Open MEEG resulting in a boundary element method head model that accounts for the brain, CSF, skin, and skull conductivity properties [45]. An MNE kernel was then implemented to generate a current source density (CSD) time series across a resultant triangular mesh (15,002 vertices) representing the cortical envelope. The vertices were parcellated into 68 cortical nodes (Nodes) according to the Desikan-Killiany (DK) atlas [46]. The DK atlas includes fourteen hierarchical anatomical grouping variables combining information across cortical regions (see S2 Fig). The temporal lobe region included homologous right and left pairs of the following nodes: banks of the superior temporal sulcus and transverse temporal gyrus, which are on or adjacent to AC.

**Time-frequency analyses.** For each DK atlas node, waveforms were analyzed using the *newtimef* function included with EEGLAB 2021. Convolution parameters included using Morlet wavelets with one Hz frequency steps using a linearly increasing cycle length from one cycle at the lowest frequency (2 Hz) to 30 cycles at the highest frequency (120 Hz). For any segmented amplitude time series and frequency step, *newtimef* function generates mean ITPC (aka phase locking value or intertrial coherence), single-trial power (STP), and event-related spectral perturbation (ERSP). A total of ten phase and power variables were generated for each subject per atlas node to account for the frequency band and time point (see below).

**Phase measures.** ITPC was examined at predefined time points corresponding to chirp frequency intervals as follows: onset (2–13 Hz), gamma1 (30–57 Hz), and gamma2 (63–100 Hz). Offset ITPC was evaluated at 2–13 Hz. ITPC estimates the degree of phase-locked responses across trials; however, it can be biased by trial number. To avoid bias, raw ITPC values were corrected for trial numbers by subtracting the critical r value according to the following equations [13]:

$$r_{critical} = \sqrt{\frac{-1}{trial\ count * \log\log\ (.5)}} \tag{1}$$

$$ITPC_{corrected} = ITPC_{raw} - r_{critical} \tag{2}$$

**Power measures.** Trial-level STP and ERSP were computed across alpha (8–12 Hz), gamma1, and gamma2 frequencies to be comparable with previous scalp electrode findings [13,14]. STP measures the spectral power of the frequency response, while ERSP quantifies the change in spectral power by frequency corrected for the pre-stimulus baseline (-500–0 ms).

**Effective connectivity.** Transfer entropy (TE) was used to study targeted directional interactions to investigate causal relationships between frontal and temporal sources. TE is an information theory-based effective connectivity measure suitable for identifying non-linear interactions and is robust to volume conduction [47]. The estimation of TE was implemented using the RTransferEntropy [48] package in R4.2. TE was based on Rényi entropy, an algorithm commonly used in neuroscience applications [49].

$$RT_{J \to I}(k,l) = \frac{1}{1-q} \log \left( \frac{\sum_i \phi_q\left(i_t^{(k)}\right) p^q\left(i_{t+1}|i_t^{(k)}\right)}{\sum_{i,j} \phi_q\left(i_t^{(k)} j_t^{(l)}\right) p^q\left(i_{t+1}|i_t^{(k)} j_t^{(l)}\right)} \right) \tag{3}$$

Pairs of artifact-free node-level epoched time series (60 trials) were used as X and Y input vectors, with the default Markov order of 3 for k and 3. To assess the statistical significance of TE, the Markov block bootstrap method (300 replications) was employed to preserve the dependencies within each time series under the null hypothesis of no information transfer [50]. A maximum p-value of p ≤ 0.01 was considered significant for a connection. The empirical distribution of the time series was discretized into 5% and 95% quantiles. After estimating TE, the percentage of trials that exhibited a significant TE connection was calculated for each subject in order to facilitate group comparison.

## Statistical analyses

**Linear modeling.** All statistical analyses were performed in R (version 4.2). To account for individual variation in electrode position, we opted to perform statistical modeling at the region level and use nodes within a DK region as replicates (S1 Table). We conducted a series of linear mixed-effects models (LME) for each of the ten response variables (see Phase and Power measures) using the NLME library with the general form:

$$y = X\beta + Zb + e \tag{4}$$

Where y is the response variable, X is the design matrix for the fixed effect coefficients, $\beta$, Z is the design matrix of random effects coefficient, b, and e is the vector of random errors. Denominator degrees of freedom for each fixed effect (and required for conditional F-tests) are estimated with NLME using the method described in Pinheiro and Bates [51]. Fixed effects included Group (FXS or TDC), Sex (male or female), and Region (n = 14; S2 Fig). After examining various link functions, ITPC response variables were log-transformed and raised to 1/3 power. This was validated using Akaike's information criterion (AIC), examining plots of the distribution of the residuals, residuals over predicted values, and quantile-quantile plots. Based on significant interactions, post-hoc contrasts between least-squared means (LSMEANS) were conducted and corrected using a 5% false discovery rate (FDR) [52]. Cohen's d effect sizes (ES) were calculated for pairwise comparisons. Though adjusted p ≤ .05 was considered statistically significant, adjusted p < .10 is presented along with ES to provide suggestive findings that may be of heuristic interest. Clinical correlations: Spearman's ρ was used to estimate the association between clinical measures and EEG response variables. We examined relationships across all FXS participants for our primary correlation analysis and applied a 10% FDR correction. We also explored uncorrected correlations within males with FXS.

## Results

### Alterations of intertrial phase coherence (ITPC) in FXS

**Reduction in gamma band ITPC in FXS.** Our findings indicate a disruption in gamma-band chirp frequency synchronization in FXS. Disruption was larger in males in the right temporal, right parietal, left prefrontal, and bilateral central regions (see Fig 2A and Table 2). A representative time-frequency plot of right temporal ITPC by group is shown in Fig 3A, with group differences in Fig 3B. For gamma1 ITPC, we found a significant interaction effect of group, sex, and region ($F_{13,4973}$ = 1.9, p < .001; see Fig 3C). For gamma2 ITPC, a two-way interaction between group and region was significant ($F_{13,4973}$ = 4.6, p < .001; see Fig 3D). Post-hoc comparisons found a significant reduction of gamma2 ITPC only in the left temporal cortex (LT: $t_{71}$ = -4.4, p = <.001, adj. p = <.001), with a trend within the right temporal cortex (RT: $t_{71}$ = -2.7, p = .009, adj. p = .065).

For each chirp-associated EEG feature, bar plots depict 5% FDR corrected post-hoc pairwise comparisons between groups of significant fixed or interaction effects (noted below the x-axis). Each significant bar (*) is annotated with the associated Cohen's d effect size. Each pair of bars represents brain regions corresponding to either left (left bar) or right (right bar) hemispheres. Positive t-values indicate that the feature estimates in FXS are greater than in matched controls. A. Intertrial Phase Coherence (ITPC): Left: Males with FXS demonstrate impairment in gamma ITPC to the auditory chirp.

# A. Intertrial Phase Coherence (ITPC) by Frequency Band

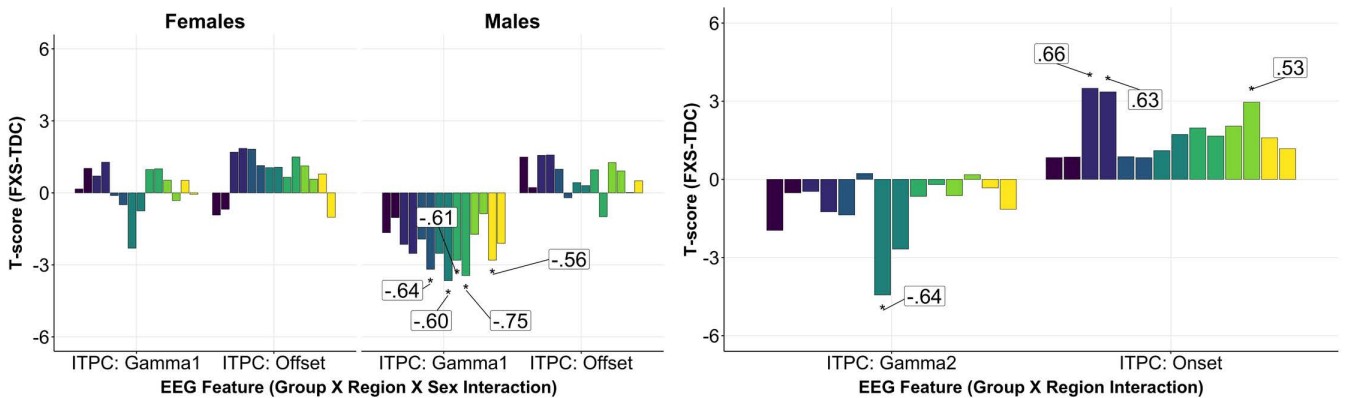

# B. Single Trial Power (STP) by by Frequency Band

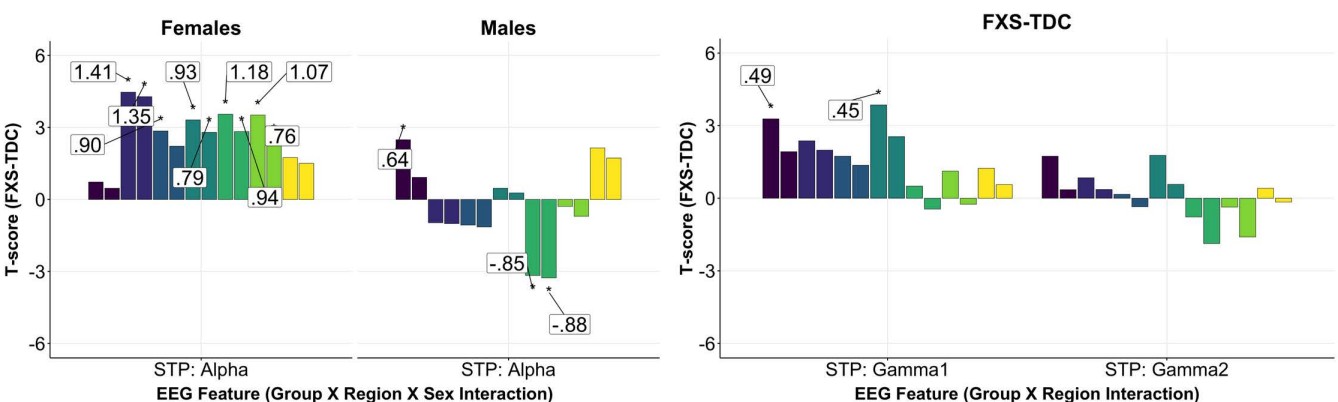

# C. Event-related Spectral Perturbation (ERSP) by Frequency Band

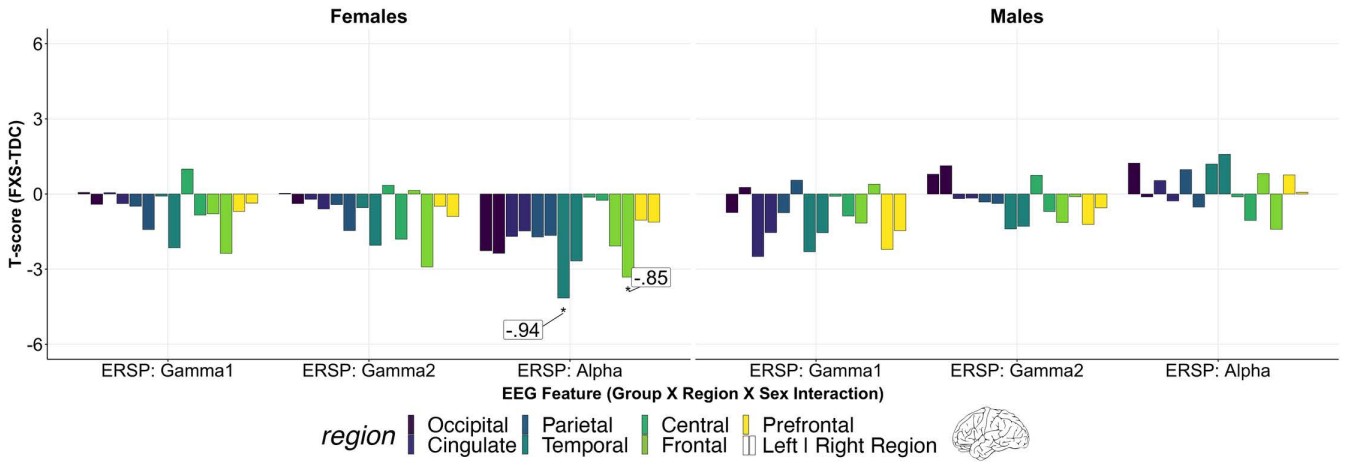

**Fig 2. Altered neural responses to auditory chirp in Fragile X Syndrome (FXS).**

**Table 2. Significant intertrial phase coherence (ITPC) contrasts by frequency band.**

| Variable | Region | Sex | FXS | TDC | DF | F | ES | p | adj. p |
|---|---|---|---|---|---|---|---|---|---|
| **ITPC: Gamma1** | LC | M | .018±.027 | .030±.031 | 71 | -2.8 | -.61 | .006 | .037 |
| | LPF | M | .006±.013 | .016±.018 | 71 | -2.8 | -.56 | .007 | .037 |
| | LT | F | .016±.015 | .028±.032 | 71 | -2.3 | -.46 | .024 | .085 |
| | LT | M | .019±.032 | .029±.037 | 71 | -2.5 | -.41 | .014 | .056 |
| | RC | M | .018±.020 | .036±.030 | 71 | -3.4 | -.75 | <.001 | .013 |
| | RL | M | .033±.038 | .045±.040 | 71 | -2.5 | -.51 | .014 | .056 |
| | RP | M | .015±.020 | .027±.022 | 71 | -3.2 | -.64 | .002 | .020 |
| | RT | M | .017±.022 | .033±.043 | 71 | -3.7 | -.60 | <.001 | .013 |
| **ITPC: Gamma2** | LT | | .008±.015 | .016±.020 | 71 | -4.4 | -.64 | <.001 | <.001 |
| | RT | | .008±.010 | .014±.020 | 71 | -2.7 | -.39 | .009 | .065 |
| **ITPC: Onset** | LL | | .105±.089 | .064±.054 | 71 | 3.5 | .66 | <.001 | .009 |
| | RF | | .074±.067 | .047±.043 | 71 | 3.0 | .53 | .004 | .019 |
| | RL | | .111±.090 | .070±.055 | 71 | 3.4 | .63 | .001 | .009 |

Significant corrected linear mixed model post-hoc group contrasts of least-squared means (± standard error) of ITPC. Abbreviations: ES, Cohen's *d* effect size; adj. p, 5% false discovery rate corrected p values; FXS, Fragile X Syndrome; TDC, typically developing controls; DF, degrees of freedom; FDR, false discovery rate; LC, left central; RC, right central; LPF, left prefrontal; LT, left temporal; RT, right temporal; RL, right cingulate; LL, left cingulate; RP, right parietal.

Right: Individuals with FXS display increased onset ITPC to the auditory chirp compared to controls localized to frontal and cingulate regions. Cingulate regions in this EEG-optimized atlas primarily refer to cingulate cortical regions and other deeper sources. B. Single trial power (STP): Left: Alpha STP is increased in FXS females and reduced in FXS males compared to controls. Right: Individuals with FXS demonstrate heightened gamma STP in the left temporal and left occipital regions. Similar trend-level effects were found in the right temporal (p = .013, adj. p = .061) and left cingulate (p = .020, adj. p = .072) regions. C: Event-related spectral perturbation (ERSP): Left: Females with FXS demonstrated a significant (or trending) increase in alpha power across bilateral cortical regions. Right: A significant group x sex x region interaction for gamma1 ERSP (p < .001) and gamma2 ERSP (p = .017) was present, but no pairwise comparisons survived correction. A review of raw p values and effect sizes suggests the interaction was driven by decreased gamma1 ERSP in males with FXS.

**Enhanced frontal onset ITPC in FXS.** Onset ITPC to the chirp stimulus was increased in FXS (significant group by region interaction; $F_{13,4973}$ = 2.9, p < .001; see Table 2) within the right frontal (RF: $t_{71}$ = 3.0, p = .004, adj. p = .019), right cingulate (RL: $t_{71}$ = 3.4, p = .001, adj. p = .009), and left cingulate regions (LL: $t_{71}$ = 3.5, p=<.001, adj. p = .009; see Fig 2A and 3D). Due to the significant decline in the precision of EEG source localization at deeper depths, the cingulate area within the EEG-optimized DK atlas includes the cingulate cortex and other deep brain sources such as the inferior colliculus (see S1 Table) [30,53].

**No offset ITPC differences were found across FXS and TDC.** Although a significant interaction effect of group x sex x region was identified in the LME of offset ITPC ($F_{13,4973}$ = 1.7, p < .047; see Fig 2), no pairwise contrasts survived correction for multiple comparisons. The effect is likely driven by increased offset ITPC in the FXS group.

## Alterations of Single Trial Power (STP)

**Elevation in asynchronous gamma single-trial power.** Elevated background gamma STP has been detected at the scalp level in humans with FXS and Fmr1[-/-] KO studies at rest and in response to the auditory chirp [13,14,54,55]. Considering deficits in gamma ITPC in FXS, this background activity has been hypothesized to represent asynchronous

## Enhanced onset and diminished interstimulus ITPC in response to auditory chirp in FXS

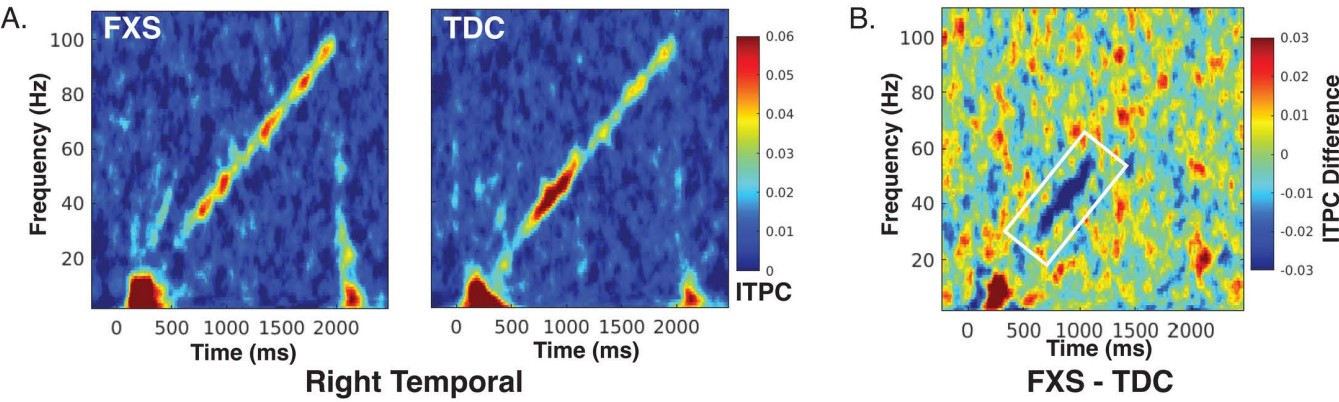

**Right Temporal**  ·  **FXS - TDC**

## ITPC to auditory chirp stimulus varies by group, region, and sex.

C.  **Significant group x sex by region interaction (5% FDR):**

ITPC: Gamma1

Male

Fem

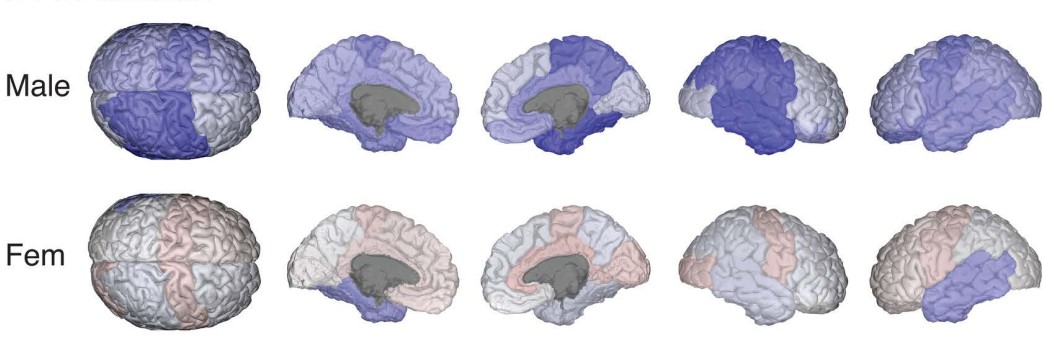

D.  **Significant group x region interaction (5% FDR):**

ITPC: Gamma2

ITPC: Onset

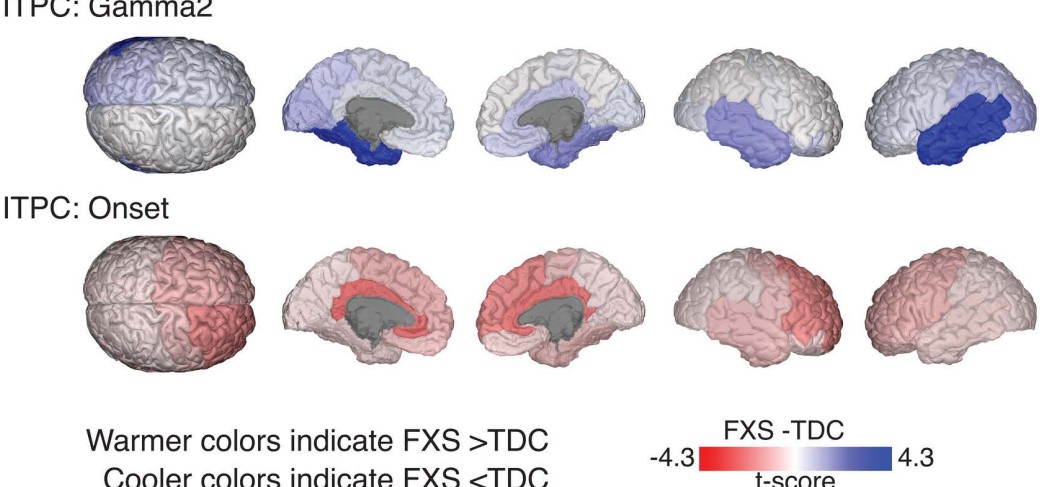

Warmer colors indicate FXS >TDC
Cooler colors indicate FXS <TDC

FXS -TDC
-4.3 ——— 4.3
t-score

**Fig 3. Intertrial phase coherence in response to auditory chirp.** A. Time-frequency heatmaps for depicting mean ITPC response by group in the right temporal region. B. ITPC difference plot (FXS-TDC) with white outline marking a significant reduction of Gamma1 ITPC in FXS. C. Brain plots depicting least-square means contrasts (t-values) of Gamma1 ITPC. D. Brain plots depicting least-square means contrasts (t-values) of Gamma2 and onset ITPC.

background noise. For gamma1 and gamma2 STP, we predicted that a significant increase in gamma STP in FXS within temporal regions with a larger effect in males. We found two significant, two-way interactions (group x sex and sex x region). Based on our a priori hypothesis, we examined pairwise contrasts for the significant group x region interaction effect (gamma1 STP: $F_{13,4973}$ = 2.6, p = .002; gamma2 STP: $F_{13,4973}$ = 2.5, p = .002). We identified significant and trend level increases in gamma STP in participants with FXS within the left temporal (LT: $t_{71}$ = 3.9, p = <.001, adj. p = .004), right temporal (RT: $t_{71}$ = 2.5, p = .013, adj. p = .061), left cingulate (LL: $t_{71}$ = 2.4, p = .020, adj. p = .072), and left occipital regions (LO: $t_{71}$ = 3.3, p = .002, adj. p = .011) (see Fig 4E, 4F). A time-frequency plot of left temporal gamma1 STP contrasts by group is shown in Fig 4A with regional differences shown in Fig 4E. No gamma2 STP pairwise contrasts survived correction.

**Alpha STP increased in FXS females and decreased in FXS males compared to controls.** For alpha STP, a significant group x region x sex interaction effect was present ($F_{13,4973}$ = 9.6, p < .001). Sex-matched contrasts of alpha STP from the left central region are depicted in Fig 4C and 4D. Males with FXS demonstrated decreased alpha STP across the central cortex (LC: $t_{71}$ = -3.2, p = .002, adj. p = .009; RC: $t_{71}$ = -3.3, p = .002, adj. p = .008; see Table 3) and increased STP in the left occipital cortex (LO: $t_{71}$ = 2.5, p = .015, adj. p = .035). Females with FXS demonstrated a significant (or trending) increase in alpha power across bilateral cortical regions, including central, frontal, cingulate, and parietal regions ($t_{71}$ = 2.1–4.3, p = <.001 to.029, adj. p. = <.001 to.07; see Table 3).

## Alterations of event related spectral perturbation (ERSP)

**Alpha ERSP is decreased in females with FXS.** For alpha ERSP, a significant group x sex x region interaction was found ($F_{13,4973}$ = 2.8, p < .001). Corrected pairwise contrasts demonstrated a significant decrease in alpha ERSP following chirp stimulus in females with FXS in the left temporal and right frontal regions (see Table 4). Visualization of t-value region differences between FXS and control females show a broader trend of decreased alpha ERSP across multiple cortical regions (see Fig 4F). A significant group x sex x region interaction was found for gamma1 ERSP ($F_{13,4973}$ = 2.7, p < .001) and gamma2 ERSP ($F_{13,4973}$ = 2.0, p = .017). No pairwise contrasts survived 5% FDR correction. LME model results for each response variable and any significant interaction effects are summarized in S2 Table. To better visualize gradients and asymmetry in ITPC response across the cortex, we also plotted region contrasts on a three-dimensional glass brain for each measure.

## Increased frontotemporal information flow in males with FXS

We next calculated Renyi transfer entropy (TE) to estimate directional information flow between the right frontal and auditory cortex regions during the onset ITPC response. We hypothesized that greater frontotemporal information flow would be present in FXS. We focused only on the right frontal region as these nodes had significantly greater onset ITPC in FXS (onset ITPC FXS-TDC: caudal middle frontal R (cMFG; t = 2.73, p = .001), pars triangularis R (pTRI, t = 2.69, p = .011), and rostral middle frontal R (rMFG; t = 2.44, p = .019)). We estimated trial-level TE between these three frontal nodes and the right and left superior temporal gyrus (STG; see S3 Fig). Significant connections were determined by bootstrap repetitions [48]. For each subject, we calculated the percentage of trials (out of 60) with significant TE pairs (referred to as TE %) for within sex statistical comparison between groups. We found TE % was increased in males with FXS from the right STG to the right cMFG (t = 2.13, p = .04; EF = .63), from the right pTRI to the right STG (t = 2.14, p = .04, EF = .63), and the right STG to the right MFG (t = 2.96; p = .005, EF = .87) (see Fig 5A–C).

Reyni's transfer entropy (TE) was calculated using source localized time series data between right frontal nodes demonstrating significant (sig.) onset intertrial phase coherence (ITPC) (cMFG, pTRI, rMFG) and the right and left STG (estimated source of the auditory cortex). Bootstrap repetitions were used to determine if a directional connection was significant (p < .01). For each subject and node pair, the % of trials demonstrating sig. TE was calculated. A. Group contrasts (within sex) of % of trials demonstrating sig. transfer entropy (TE) by frontotemporal node pairing between FXS and

**Elevation in single-trial gamma power in FXS and increase alpha ERSP in females with FXS**

A. Gamma STP: FXS > TDC
B. Alpha STP: Sex Effect
C.
D. Alpha ERSP: $FXS_{females}-TDC_{females}$

FXS > TDC; 5% FDR p < .05
Left Temporal

FXS - TDC (Males)
FXS - TDC (Females)
5% FDR p < .05
Left Central

FXS(F) < TDC(F); 5% FDR p < .01
Right Frontal

**STP and ERSP to auditory chirp stimulus varies by group, region, and sex.**

E. **Significant group × region interaction (5% FDR):**

STP: Gamma1

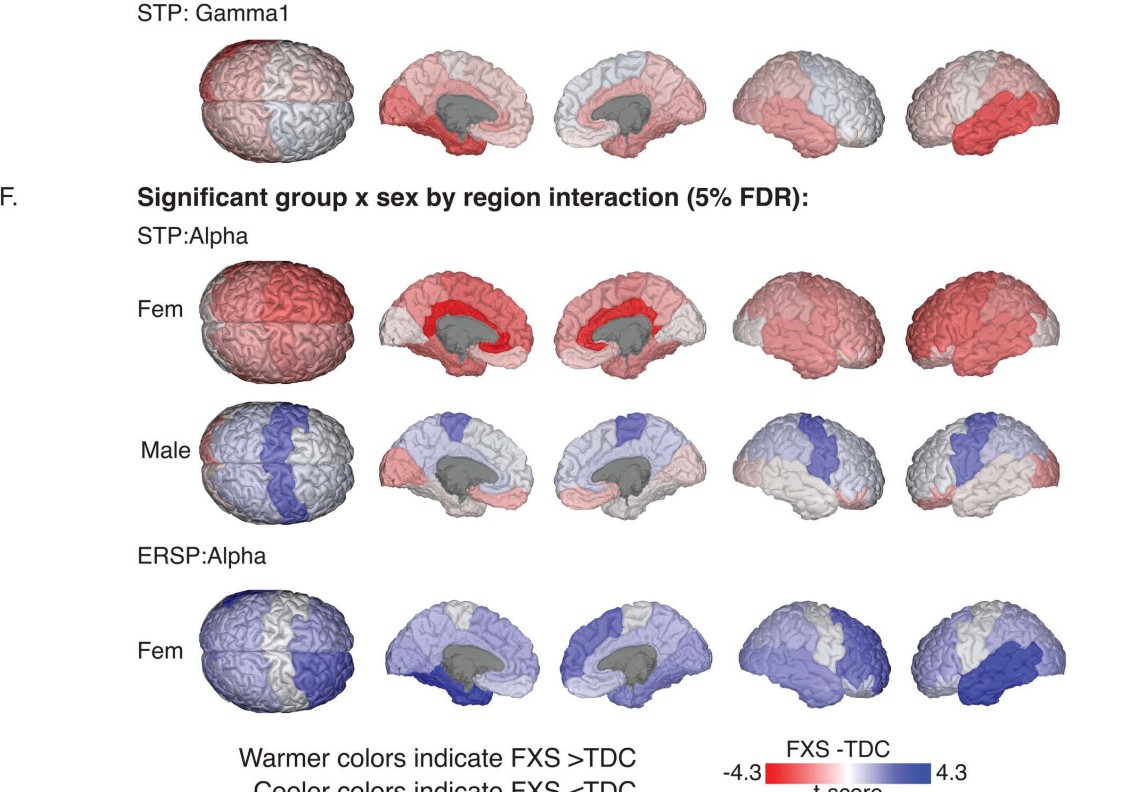

F. **Significant group × sex by region interaction (5% FDR):**

STP:Alpha

Fem

Male

ERSP:Alpha

Fem

Warmer colors indicate FXS >TDC
Cooler colors indicate FXS <TDC

FXS -TDC
-4.3 ▬ 4.3
t-score

**Fig 4. Single-trial power (STP) and event-related spectral perturbation (ERSP) in response to auditory chirp.** A-D. Exemplar time frequency heatmaps visualizing STP and ERSP contrasts (FXS-TDC) which are associated with significant linear mixed model contrasts. E-F. Brain visualizations depicting least-square means contrasts (t-values) of STP and ERSP of significant interaction effects.

typically developing controls (TDC). Bar plots represent t-score (FXS-TDC) with positive values indicating greater % of trials with sig. TE in the FXS group, while negative values indicate a greater % of trials with sig. TE in the control group. B. Subject-level jitter plot superimposed on boxplots of % of trials with sig. TE across the three sig. frontotemporal pairs. C. Density plot of % of trials with sig. TE across the three sig. frontotemporal pairs. Abbreviations: cMFG, caudal middle

**Table 3. Significant asynchronous single-trial power (STP) contrasts by frequency band.**

| Variable | Region | Sex | FXS | TDC | DF | F | ES | p | adj. p |
|---|---|---|---|---|---|---|---|---|---|
| **STP: Gamma1** | LL | | -207.3±3.0 | -208.7±3.1 | 71 | 2.4 | .36 | .020 | .072 |
| | LO | | -200.6±4.2 | -202.6±4.0 | 71 | 3.3 | .49 | .002 | .011 |
| | LT | | -199.0±4.0 | -201.0±3.7 | 71 | 3.9 | .45 | <.001 | .004 |
| | RT | | -199.8±4.0 | -201.2±3.7 | 71 | 2.5 | .30 | .013 | .061 |
| **STP: Alpha** | LC | F | -193.3±4.9 | -196.7±3.7 | 71 | 3.6 | 1.18 | <.001 | .005 |
| | LC | M | -196.2±2.5 | -193.7±2.9 | 71 | -3.2 | -.85 | .002 | .009 |
| | LF | F | -194.7±4.2 | -197.8±3.0 | 71 | 3.5 | 1.07 | <.001 | .005 |
| | LL | F | -195.1±4.1 | -199.2±2.8 | 71 | 4.5 | 1.41 | <.001 | <.001 |
| | LO | M | -193.5±3.2 | -195.3±3.7 | 71 | 2.5 | .64 | .015 | .035 |
| | LP | F | -196.5±4.5 | -199.1±4.4 | 71 | 2.9 | .90 | .006 | .019 |
| | LPF | M | -189.9±4.1 | -191.4±4.0 | 71 | 2.1 | .55 | .035 | .070 |
| | LT | F | -192.5±4.0 | -195.2±3.8 | 71 | 3.3 | .93 | .001 | .008 |
| | RC | F | -193.3±3.8 | -196.1±3.0 | 71 | 2.8 | .94 | .006 | .019 |
| | RC | M | -196.0±2.8 | -193.4±3.4 | 71 | -3.3 | -.88 | .002 | .008 |
| | RF | F | -195.4±4.2 | -197.6±2.9 | 71 | 2.5 | .76 | .015 | .035 |
| | RL | F | -195.4±4.0 | -199.3±2.7 | 71 | 4.3 | 1.35 | <.001 | <.001 |
| | RP | F | -196.0±3.4 | -198.0±3.5 | 71 | 2.2 | .70 | .029 | .064 |
| | RT | F | -193.3±4.3 | -195.6±4.0 | 71 | 2.8 | .79 | .007 | .019 |

Significant corrected linear mixed model post-hoc contrasts of least-squared means following correction of STP. Abbreviations: ES, Cohen's *d* effect size; adj. p, 5% false discovery rate corrected p values; FXS, Fragile X Syndrome; TDC, typically developing controls; DF, degrees of freedom; FDR, false discovery rate; LF, left frontal; RF, right frontal; LC, left central; RC, right central; LPF, left prefrontal; LT, left temporal; RT, right temporal; LL, left cingulate; RL, right cingulate; LP, left parietal; RP, right parietal; LO, left occipital.

**Table 4. Significant event-related spectral perturbation (ERSP) contrasts.**

| Variable | Region | Sex | FXS | TDC | DF | F | ES | p | adj. p |
|---|---|---|---|---|---|---|---|---|---|
| **ERSP: Alpha** | LT | F | -.32±.40 | -.03±.36 | 71 | -4.2 | -.94 | <.001 | .003 |
| | RF | F | -.22±.40 | .03±.36 | 71 | -3.3 | -.85 | .001 | .020 |
| | RT | F | -.26±.32 | -.07±.42 | 71 | -2.7 | -.61 | .009 | .088 |

Significant corrected linear mixed model post-hoc contrasts of least-squared means following correction of ERSP. Abbreviations: ES, Cohen's *d* effect size; adj. p, 5% false discovery rate corrected p values; FXS, Fragile X Syndrome; TDC, typically developing controls; DF, degrees of freedom; FDR, false discovery rate; RF, right frontal; LT, left temporal; RT, right temporal.

frontal; pTRI, pars triangularis; rMFG, rostral middle frontal; STG, superior temporal gyrus; R, right; L, left. Significance: *, p<.05; **, p<.01. For all boxplots, the center line represents the median, the lower and upper hinges correspond to the 25th and 75th percentiles, and the upper and lower whiskers extend from the hinge to the largest or smallest value, respectively, no further than 1.5 * distance between the 25th and 75th percentiles.

## Clinical correlations

For all FXS participants, we examined 10% FDR corrected Spearman's correlations between bilateral frontal and temporal regions with significant EEG alterations with 13 clinical measures (780 correlations; see Table 5). As full mutation males with FXS are expected to have absent or very low levels of FMRP expression, we also explored uncorrected clinical correlations in an FXS male-only subgroup (see Fig 6A–D and Table 6).

Neural synchronization and background gamma power are associated with clinical measures

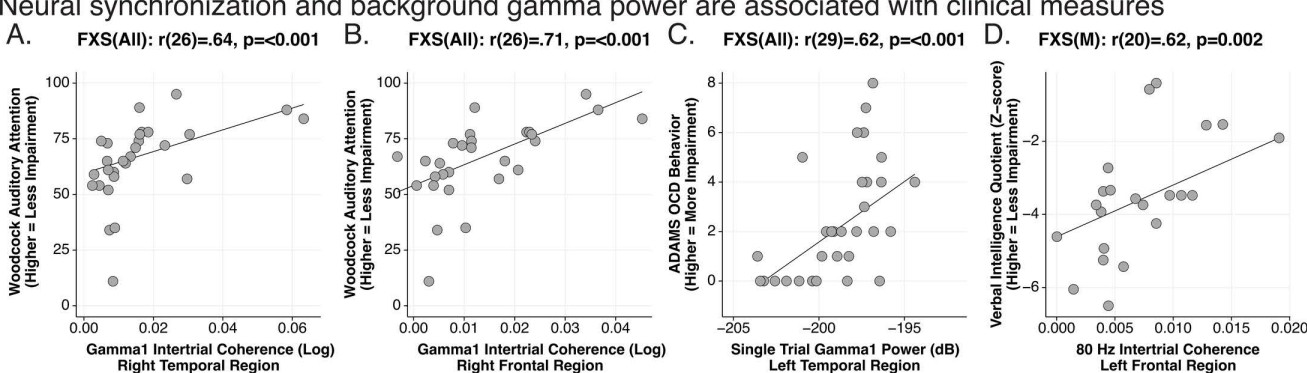

Synchronization-asynchronous ratio as an estimate of neural signal to noise in FXS

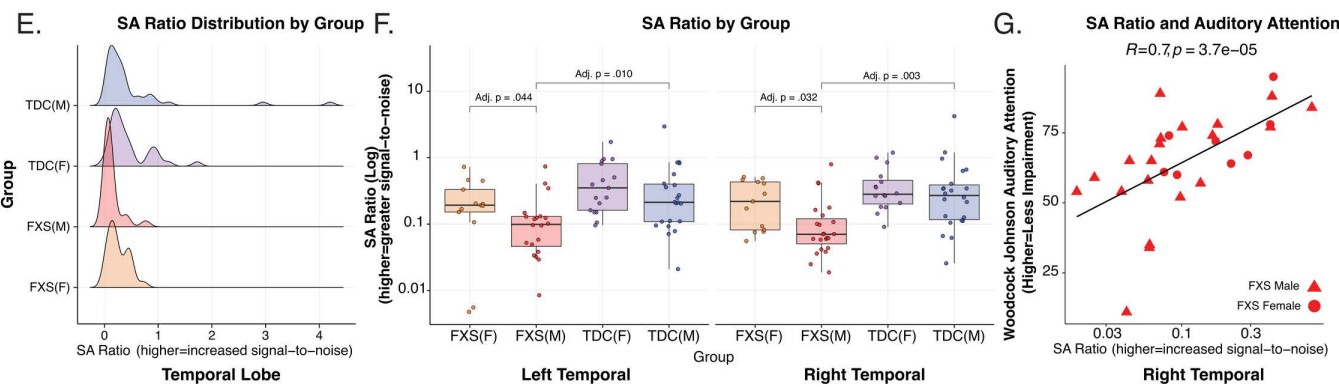

**Fig 5. Increased frontotemporal information flow in males with Fragile X Syndrome.**

**Table 5. Corrected clinical correlations with EEG features across all FXS participants.**

| Measure | Feature | n | rho | p | 10% FDR |
|---------|---------|-----|------|--------|---------|
| ADAMS OCD | STP: Left Temporal Gamma1 | 31 | .62 | .0002 | .0355 |
| ADAMS OCD | STP: Left Temporal Gamma2 | 31 | .64 | .0001 | .0355 |
| SCQ Total | ITPC: Left Frontal Gamma1 | 31 | -.58 | .0007 | .0838 |
| SCQ Total | STP: Right Temporal Gamma2 | 31 | .57 | .0008 | .0838 |
| WJ3 | ITPC: Right Frontal Gamma1 | 28 | .71 | <.0001 | .0167 |
| WJ3 | ITPC: Right Temporal Gamma1 | 28 | .64 | .0002 | .0355 |
| WJ3 | ITPC: Right Temporal gamma2 | 28 | .65 | .0002 | .0355 |

Abbreviations: FXS, Fragile X Syndrome; SCQ, Social Communication Questionnaire; WJ-3, Woodcock III Tests of Cognitive Abilities; ADAMS, Anxiety, Depression, and Mood Scale.

**Auditory attention.** All FXS: The WJ3 quantifies selective auditory attention (including speech-sound discrimination and resistance to auditory stimulus distraction), with higher scores indicating better performance [36]. The task can be reliably performed in children and those with intellectual disability. We found a positive correlation between WJ3 auditory attention task performance and phase locking with Gamma1 ITPC (right temporal: rho(27)=.64; p<.001, adj. p=.035; see Fig 6A; right frontal: rho(27)=.71; p<.001, adj. p=.017; see Fig 6B) and gamma2 ITPC (right temporal: rho(27)=.65;

**Frontotemporal transfer entropy (TE) is increased in males with FXS**

**A.  Comparison of Frontotemporal TE by Group and Sex**

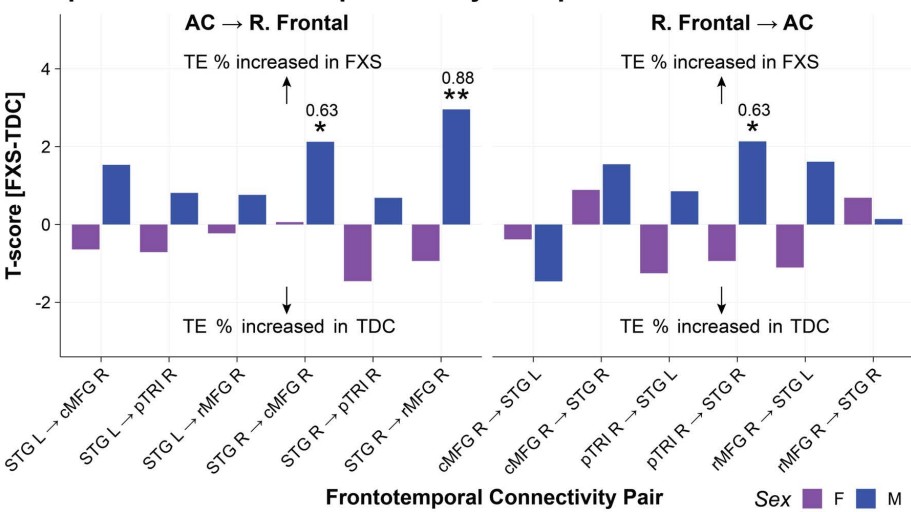

**B.  Subject-level sig. TE %**   **C.  Distribution of sig.TE pairs**

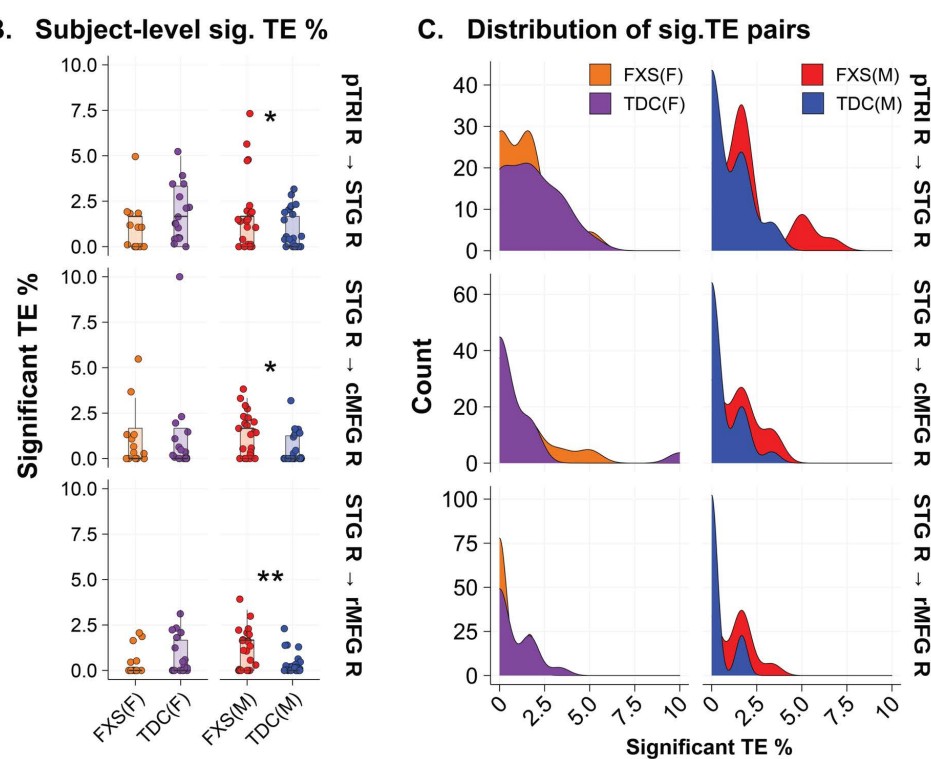

**Fig 6.  Clinical Correlations and Neurobehavioral Biomarker.** Neural synchronization and background gamma power are associated with clinical measures. A-D. Exemplar scatter plots of clinical correlations with EEG variables across all Fragile X Syndrome (FXS) subjects, and a subgroup of full mutation males are shown. Scatterplots in each box depict subject-level bivariate Spearman's correlations. E-G. Synchronization-asynchronization ratio (SA ratio) estimates neural signal to noise (SNR) in FXS. The SA ratio is a composite measure of the degree of gamma1 neural synchronization ("phase locking") as a ratio of background gamma power ("noise"). E. Distribution of SA in males with FXS is skewed towards a lower SA ratio (reduced SNR). F. Boxplots with superimposed scatter plot of SA ratio: Data points represent the average SA ratio within the right and left temporal region per subject. SA ratio in males with FXS is significantly reduced in bilateral temporal regions compared to either control males or females with FXS (see Table 7). For all boxplots, the center line represents the median, the lower and upper hinges correspond to the 25th and 75th percentiles), and the upper and lower whiskers extend from the hinge to the largest or smallest value, respectively, no further than 1.5 times the distance between the 25th and 75th percentiles. G. Temporal SA ratio is highly correlated with auditory attention, so greater SNR is associated with better auditory attention performance.

**Table 6. Uncorrected clinical correlations with chirp EEG features across the subgroup of males with FXS.**

| Measure | Feature | n | rho | p |
|---|---|---|---|---|
| ABC FXS: Lethargy/social withdrawal | ITPC: Offset(RT) | 20 | -.49 | .0266 |
| ABC FXS: Hyperactivity/Noncompliance | ITPC: gamma2(LF) | 20 | -.53 | .0174 |
| ABC FXS: Inappropriate speech | ERSP: Gamma2(RT) | 20 | -.49 | .0298 |
| | ITPC: Onset(LF) | 20 | .49 | .0284 |
| | ITPC: Onset(RF) | 20 | .55 | .0128 |
| | ITPC: Onset(RT) | 20 | .47 | .0368 |
| ADAMS General Anxiety | ERSP: Gamma1(LO) | 21 | .45 | .0421 |
| | ITPC: Onset(RT) | 21 | .50 | .0224 |
| ADAMS Obsessive/Compliance Behavior | ERSP: Gamma1(RF) | 21 | -.46 | .0367 |
| | ITPC: Offset(RT) | 21 | -.51 | .0173 |
| Age at Visit | ERSP: Alpha(LF) | 23 | -.48 | .0198 |
| | ITPC: Gamma1(RO) | 23 | .46 | .0263 |
| | ITPC: Gamma1(RT) | 23 | .43 | .0412 |
| Deviation IQ | ITPC: Gamma1(LT) | 22 | .44 | .0423 |
| NVIQ Z Score | ITPC: Gamma1(LT) | 22 | .48 | .0244 |
| VIQ Z Score | ITPC: Gamma1(LF) | 22 | .45 | .0348 |
| | ITPC: Gamma2(LF) | 22 | .62 | .0021 |
| WJ3 | ERSP: Gamma2(LT) | 20 | .50 | .0253 |
| | ERSP: Gamma2(RO) | 20 | -.47 | .0373 |
| | ITPC: Gamma1(LO) | 20 | .52 | .0193 |
| | ITPC: Gamma1(LT) | 20 | .52 | .0175 |
| | ITPC: Gamma1(RF) | 20 | .72 | .0003 |
| | ITPC: Gamma1(RO) | 20 | .55 | .0118 |
| | ITPC: Gamma1(RT) | 20 | .65 | .0017 |
| | ITPC: Gamma2(RT) | 20 | .60 | .0054 |

Abbreviations: FXS, Fragile X Syndrome; NVIQ, Non-verbal intelligence scale; VIQ, verbal intelligence scale; SCQ, Social Communication Question-naire; ABC, Aberrant Behavior Checklist; WJ-3, Woodcock III Tests of Cognitive Abilities; ADAMS, Anxiety, Depression, and Mood Scale.

p < .001, adj. p = .035). FXS Males: This association of gamma1 ITPC and WJ3 was preserved in the subgroup of males with FXS (see Table 6).

**Other clinical measures.** All FXS: Higher (worse) Anxiety, Depression, and Mood Scale (ADAMS) obsessive severity scores were associated with increased left temporal gamma1 (left temporal: rho(30)=.62; p < .001, 5% FDR p = .035; see Fig 6C) and gamma2 STP (left temporal: rho(30)=.64; p < .001, 5% FDR p = .035). We found trend level associations (10% FDR) such that decreased gamma1 ITPC (left frontal: rho(30)=-.58; p < .001, 5% FDR p = .083) and increased gamma2 STP (right temporal: rho(30)=.64; p < .001, 5% FDR p = .083) were associated with SCQ (social communication impairment). In FXS males (see Table 6), higher left temporal gamma1 ITPC was associated with higher non-verbal and full-scale IQ scores. Increased left frontal gamma1 and gamma2 ITPC was correlated with increased verbal IQ (see Fig 6D). Onset ITPC of bilateral frontal and right temporal regions were associated with ABC measures of inappropriate speech. Higher levels of gamma1 and gamma2 ERSP were associated with worsening ADAMS anxiety scores, OCD scale scores and ABC inappropriate speech. ERSP alpha power was inversely related to age at visit.

**Transfer entropy.** Uncorrected Spearman correlation of TE % and cognitive measures within males with FXS revealed a moderate, negative correlation between TE % of the pTRI R to the STG R and intelligence (FSIQ: r = -0.610, p = 0.003,

n = 22) and verbal z-score (r = -0.497, p = 0.02, n = 22). This negative correlation suggests that higher frontotemporal information flow in these specific brain regions may be associated with lower cognitive performance in males with FXS.

Gamma synchronous-asynchronous (SA) ratio suggests reduced signal-to-noise ratio of auditory processing in FXS Individuals with FXS had increased background gamma1 STP, though the gamma1 ITPC deficit was primarily present in males. We hypothesized that the relationship between gamma phase synchrony (gamma ITPC) and background elevation in unsynchronized gamma power (STP) would be an indicator of signal-to-noise ratio (SNR), with lower signal-to-noise ratio being associated with hyperacusis [56]. We calculated the ratio of gamma1 ITPC and gamma1 STP to derive a synchronous-to-asynchronous ratio (SA ratio). Gamma1 STP values were normalized between 0 and the maximum of gamma1 ITPC (.0164) to ensure similar scales between measures. A higher SA ratio indicates more stimulus phase locking ("signal") relative to background gamma power ("noise") (see Eq. 5).

$$synchronous - to - asynchronous_{ratio} = \frac{gamma1\ ITPC}{gamma1\ STP}$$

(5)

We focused on the SA ratio within the temporal lobes where ITPC and STP impairment was detected and the location of AC. We calculated the average SA ratio within the left and right temporal lobes for each subject (see Fig 6E). A Wilcoxon Signed-Ranks Test indicated that the SA ratio was diminished in males with FXS compared to females with FXS and control participants (see Fig 6F and Table 7). Highlighting the functional importance of this SNR index, we observed a significant positive correlation between the SA ratio in auditory cortex and WJ3 auditory attention task performance across FXS participants (R = .70, p = 3.7e-5; see Fig 6G and Table 7).

## Discussion

The mechanisms underlying auditory hypersensitivity in FXS have not yet been established. Understanding these mechanisms has broad implications as sensory alterations are impairing and highly prevalent across neurodevelopmental disorders [57–60]. Using a well-powered case-control study, we examined the spatiotemporal characteristics of neural response to the chirp stimulus. Our objective was to characterize the spatiotemporal characteristics of gamma abnormalities associated with auditory processing of the chirp stimulus in FXS. Our major findings in FXS include 1) a marked

**Table 7. The synchronous-to-asynchronous (SA) ratio was developed as a proposed signal-to-noise biomarker to estimate the efficiency of auditory processing in FXS.**

| Region | Side | g1 | g2 | (g1-g2) | 95% CI | W | ES | p | Adj. p |
|---|---|---|---|---|---|---|---|---|---|
| Temporal | Left | FXS(M) | TDC(F) | -1.66 | [-3.26 -.81] | 43 | .61 | 7.4e-06 | 4.4e-05 |
| | | FXS(M) | TDC(M) | -.89 | [-1.83 -.42] | 118 | .49 | 2.0e-03 | 9.0e-03 |
| | | FXS(M) | FXS(F) | -.80 | [.28 1.41] | 227 | .42 | 0.01 | 0.039 |
| | | FXS(F) | TDC(F) | -1.04 | [-2.92.18] | 70 | .25 | 0.094 | 0.284 |
| | | FXS(F) | TDC(M) | -.26 | [-1.44.66] | 121 | .11 | 0.468 | 0.468 |
| | | TDC(F) | TDC(M) | .62 | [-.36 2.16] | 236 | .16 | 0.172 | 0.344 |
| | Right | FXS(M) | TDC(F) | -1.45 | [-2.19 -.79] | 66 | .61 | 2.2e-04 | 1.0e-03 |
| | | FXS(M) | TDC(M) | -1.24 | [-1.96 -.37] | 105 | .49 | 5.4e-04 | 3.0e-03 |
| | | FXS(M) | FXS(F) | -.75 | [.12 2.31] | 232 | .42 | 6.0e-03 | 0.023 |
| | | FXS(F) | TDC(F) | -.61 | [-1.68.80] | 88 | .25 | 0.363 | 1 |
| | | FXS(F) | TDC(M) | -.25 | [-1.53.84] | 127 | .11 | 0.601 | 1 |
| | | TDC(F) | TDC(M) | .25 | [-.91 1.24] | 206 | .16 | 0.604 | 1 |

Abbreviations: g1, Group 1; g2, Group 2; FXS, Fragile X Syndrome; TDC, Typically Developing Controls; CI, confidence interval; W, Wilcoxon Statistic, ES, Wilcoxon effect size; Adj. p, Bonferroni-Holm corrected adjusted p values.

increase in onset phase synchronization in frontal regions, 2) reduced chirp synchronization and increased background activity in temporal regions, 3) increased frontotemporal information exchange, 4) mediation by sex for these key alterations, and 5) robust associations between clinical measures and EEG changes. These systems-level alterations may serve as biomarkers of target engagement for treatments aimed at reducing hyperarousal and improving regulation of sensory systems. The results create opportunities for back-translation by testing circuit-level mechanisms of the observed functional changes in *Fmr1*[-/-] knockout mouse models [54,61].

### Frontal region hyperresponsiveness and frontotemporal connectivity

The frontal regions of individuals with FXS showed a heightened onset ITPC response (157–165% vs. controls) to the chirp stimulus. In males with FXS, this increase in onset ITPC was associated with worsening anxiety scores. While it is typical for frontal and temporal regions to reciprocate activity in normative populations during early auditory processing [62–64], our results suggest that individuals with FXS have exaggerated salience or orientation to sensory stimulation in FXS, which could contribute to heightened anxiety levels. These functional changes may be related to alterations in frontal structure and maturation as observed from neuroimaging studies on FXS [3,4,65–68]. Through effective connectivity analysis, we discovered an increase in reciprocal information flow between frontotemporal regions. The auditory pathway from the brainstem to the cortex, contains a high-density of FMRP-expressing neurons [69], making it vulnerable to FMRP loss [70,71]. The increase in information flow was negatively correlated with cognitive function. These findings suggest that there may be a breakdown in the top-down regulatory processes responsible for filtering out irrelevant sensory information in FXS. Consequently, individuals with FXS might have an over-responsiveness of the auditory cortex to incoming auditory stimuli contributing to hyperacusis and anxiety observed [24,25]. The observed variation in frontal input may also explain the heterogeneity in auditory sensitivity observed clinically in FXS - some individuals with FXS display distress to certain noises while others may enjoy loud music [57]. These alterations in frontotemporal connectivity suggest that the integration of bottom-up sensory and top-down frontal processes influences whether sounds are perceived as pleasurable or aversive in context.

### Localization of gamma synchronization and background power alterations in FXS

Control subjects typically demonstrate strong phase synchronization to the auditory chirp stimulus in the gamma frequency band (~40 Hz), which allows for effective encoding and selective enhancement of the stimulus over background noise [7]. In individuals with FXS, we observed significant impairments in gamma phase alignment and an increase in asynchronous background gamma power in the temporal lobe regions. These findings were strongly associated with obsessive behavior, as well as alterations in social communication and auditory attention. In males with FXS, deficits in gamma synchronization in frontal regions were linked to greater reductions in verbal intelligence, auditory attention, and hyperactivity/impulsivity. By similar findings with trend-level significance after correction for multiple comparisons (e.g., effects significant at 10% but not 5% FDR), a more comprehensive understanding of abnormalities in the FXS functional auditory system emerges. Alterations in synchronization in FXS appear to occur in both bilateral temporal and prefrontal cortex, right parietal cortex, and cingulate sources, all of which play roles in auditory processing [64,72–74]. This suggests dysfunction in filtering irrelevant auditory information and maintaining an optimal excitatory-inhibitory balance. Interestingly, though excitatory-inhibitory balance may be altered, we speculate that the auditory system may otherwise have reached an "imperfect homeostasis" with basic hearing functions maintained, but struggling to cope with more complex demands and lacking the ability to compensate, leading to hypersensitivity and startle responses [60,75].

### Event-related gamma activity in FXS is elevated and stimulus-independent

How do we best reconcile 1) reduced gamma synchronization, 2) increased gamma background power, and 3) no change in gamma stimulus evoked power in FXS during the chirp? Though we previously reported these findings in scalp recordings of the chirp response, we have now effectively disentangled temporal and frontal contributions to these alterations.

The temporal lobe findings are reminiscent of *Fmr1⁻ᐟ⁻* KO brain slice microcircuits experiments in which gamma oscillations exhibit hyper-synchrony that is stimulus independent and not responsive to phase reset [21]. For example, disordered and incongruent firing patterns are observed in extracellular recordings from layer 4 of the *Fmr1⁻ᐟ⁻* KO somatosensory cortex [76,77] and auditory cortex [21]. The SA ratio captured the relationship between neural synchronization and background asynchronous activity of gamma activity within the temporal regions. We speculate that in FXS, engagement of cortical circuitry via the input sound leads to increased activity (i.e., gamma STP) but does not lead to the expected phase reset of gamma oscillations in relation to the frequency of sensory input (e.g., gamma1 ITPC). We found significantly lower SA ratios in males with FXS. Across all FXS subjects, the high degree of association between increased SA ratios and increased auditory attention performance highlights the clinical significance of our findings.

### Observed sex differences may reveal compensatory action of alpha

Our findings reveal inherent inefficiencies in temporal lobe auditory processing in Fragile X Syndrome (FXS), with sex differences. In males, reduced gamma phase synchronization and elevated asynchronous gamma power in temporal regions may indicate impairments associated with basic sensory function [78,79]. In contrast, females with FXS showed intact temporal synchronization and gamma power, along with higher signal-to-noise ratio performance, suggesting relatively spared auditory processing. While auditory hypersensitivity is common in FXS [57], the degree of sensory distress differs from person to person and is not universal and may be explained by variations in the distribution and level of FMRP expression in the auditory system) [69]. Interestingly, we observed markedly increased alpha power in females with FXS localized to deeper brain structures (cingulate or other regions such as the inferior colliculus) [80]. It is well-established that alpha power is significantly reduced in males with FXS [15,16,81–83] and proposed to represent thalamocortical dysfunction [16,84]. When considered in the context of alpha canonical role in functional inhibition, it may be that intact alpha generation in females may mitigate hyperexcitability impairments by gating information flow [85–88]. For example, when a stimulus is presented against background noise in controls, alpha power is increased in occipital and temporal regions [86,89]. Elucidating potential protective factors in females, including enhancement of alpha oscillations, could inform the development of novel therapies targeting sensory hypersensitivity across both sexes in FXS.

### Limitations

Certain limitations of the current study should be taken into when interpreting the results. First, the study only included FXS and control groups. It would be advantageous to include control groups with consist of individuals who are IQ matched with other neurodevelopmental conditions to determine the specificity of our observations to FXS. Second, increasing the number of females and targeting specific age groups would enhance understanding of sex and developmental effects. Third, as excluding medicated patients would disproportionately reduce the representation of individuals with more severe clinical presentations, we opted to include such cases. While we reported no differences in chirp response between medicated and non-medicated patients [14], current medication treatments might impact the reported findings. We excluded individuals receiving antiepileptic medications, which are known to affect EEG measures. In addition to these clinical considerations, the current findings are specific to the auditory system, and it is uncertain if the findings will apply to other modalities. Due to a lack of individual anatomical data, we utilized a standardized anatomical atlas (ICBM152) to compute the electric lead field for the EEG source modeling. While individual MRI scans can improve accuracy, prior studies have shown that EEG source estimates have sufficient spatial resolution to justify forgoing individual anatomy in certain contexts [40,90]. In addition, we examined cortical regions and used atlas nodes as replicates, taking a conservative approach with regards to the spatial precision of our analysis. There remain practical challenges with quantifying sensory hypersensitivity in individuals with intellectual disability by self- or caregiver report [91]. Instead, we used an alternative strategy by using an auditory attention task that includes both top-down cognitive processing and sensory perception that offers a more objective and feasible means of assessment applicable to participants of varying functional levels [36].

## Conclusion

Our study demonstrates distinct spatial and behavioral correlates of abnormal auditory processing in FXS. The functional auditory system in FXS displays differences by region and has altered heuristics in detecting signal to noise. Males with FXS display excessive frontal activation upon hearing a stimulus, which suggests a disruption in the modulatory influence of top-down pathway from the frontal to the auditory cortex in influencing auditory processing. Across males and females with FXS, asynchronous gamma activity is elevated and rigid, which may constrain the function of the auditory system. The findings of this study are significant for back translation, as they offer more precise electrophysiological phenotypes to target in future research utilizing region-selective or cell-selective conditional knock-out. They also have implications for therapeutic targets in locally rigid circuits and are relevant for signal-to-noise conceptual frameworks in understanding apparent paradoxical neural activity in neurodevelopmental disorders.

## Supporting information

**S1 Table. Assignment of cortical nodes to region groupings as attributed by the Desikan-Killiany (DK) atlas.** Includes MNI coordinates and vertex counts.
(DOCX)

**S2 Table. Summary of linear mixed model results for power and phase response variables.**
(DOCX)

**S1 Fig. Broadband auditory chirp stimulus.** A. Amplitude-modulated pink noise carrier waveform. B. Waterfall plot of broadband power spectrum.
(DOCX)

**S2 Fig. Atlas Regions.** Depicts 14 hierarchical regions of the Desikan-Killiany atlas encompassing 68 cortical nodes.
(DOCX)

**S3 Fig. Frontotemporal atlas nodes for transfer entropy estimation.** Shows nodes used for Renyi transfer entropy between frontal and superior temporal gyrus regions.
(DOCX)

## Acknowledgments

Dr. Sweeney passed away before the submission of the final version of this manuscript, Dr. Pedapati accepts responsibility for the integrity and validity of the data collected and analyzed. Dr. Sweeney played an integral role in the conception and development of the ideas presented.

## Author contributions

**Conceptualization:** Ernest V. Pedapati, Lauren E. Ethridge, John A. Sweeney, Khaleel Razak, Craig A. Erickson.

**Data curation:** Ernest V. Pedapati, Lauren E. Ethridge.

**Formal analysis:** Ernest V. Pedapati, Lauren E. Ethridge, Yanchen Liu, Makoto Miyakoshi, Paul S. Horn.

**Funding acquisition:** Ernest V. Pedapati, Lauren E. Ethridge, John A. Sweeney, Khaleel Razak, Devin Binder, Craig A. Erickson.

**Investigation:** Ernest V. Pedapati, Lauren E. Ethridge, Lisa A. DeStefano, Craig A. Erickson.

**Methodology:** Ernest V. Pedapati, Lauren E. Ethridge, Yanchen Liu, Lisa A. DeStefano, Makoto Miyakoshi.

**Project administration:** Ernest V. Pedapati, Lauren E. Ethridge.

**Resources:** Ernest V. Pedapati, Lauren E. Ethridge.

**Software:** Ernest V. Pedapati, Lauren E. Ethridge, Yanchen Liu.

**Supervision:** Ernest V. Pedapati, Lauren E. Ethridge, Craig A. Erickson.

**Validation:** Ernest V. Pedapati, Lauren E. Ethridge.

**Visualization:** Ernest V. Pedapati, Lauren E. Ethridge, Yanchen Liu.

**Writing – original draft:** Ernest V. Pedapati, Lauren E. Ethridge.

**Writing – review & editing:** Ernest V. Pedapati, Lauren E. Ethridge, Yanchen Liu, Rui Liu, John A. Sweeney, Lisa A. DeStefano, Makoto Miyakoshi, Khaleel Razak, Lauren M. Schmitt, David R. Moore, Donald L. Gilbert, Steve W. Wu, Elizabeth Smith, Rebecca C. Shaffer, Kelli C. Dominick, Paul S. Horn, Devin Binder, Craig A. Erickson.

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
