## [Decision Letter · Decision Letter 0]

20 Jan 2025

PONE-D-24-20546Frontal Cortex Hyperactivation and Gamma Desynchrony in Fragile X Syndrome: Correlates of Auditory HypersensitivityPLOS ONE

Dear Dr. Pedapati,

Thank you for submitting your manuscript to PLOS ONE. After careful consideration, we feel that it has merit but does not fully meet PLOS ONE’s publication criteria as it currently stands. Therefore, we invite you to submit a revised version of the manuscript that addresses the points raised during the review process.

We look forward to receiving your revised manuscript.

Kind regards,

Abdolvahed Narmashiri

Academic Editor

PLOS ONE

**Journal Requirements:**

The present study was federally funded by the National Institutes of Health (NIH)

Fragile X Center (U54HD082008 and U54HD104461) (PI: CE)

3. Please note that your Data Availability Statement is currently missing the direct link to access each database. If your manuscript is accepted for publication, you will be asked to provide these details on a very short timeline. We therefore suggest that you provide this information now, though we will not hold up the peer review process if you are unable.

5. We notice that your supplementary figures and tables are included in the manuscript file. Please remove them and upload them with the file type 'Supporting Information'. Please ensure that each Supporting Information file has a legend listed in the manuscript after the references list.

Reviewers' comments:

Reviewer's Responses to Questions

**Comments to the Author**

1. Is the manuscript technically sound, and do the data support the conclusions?

Reviewer #1: Yes

Reviewer #2: Yes

2. Has the statistical analysis been performed appropriately and rigorously? 

Reviewer #1: Yes

Reviewer #2: Yes

3. Have the authors made all data underlying the findings in their manuscript fully available?

Reviewer #1: Yes

Reviewer #2: Yes

4. Is the manuscript presented in an intelligible fashion and written in standard English?

Reviewer #1: Yes

Reviewer #2: Yes

5. Review Comments to the Author

**Reviewer #1: ** The authors observed a decrease in typical gamma phase synchrony and an increase in asynchronous gamma power in multiple regions, most strongly in the temporal cortex, in individuals with FXS during chirp stimulation. This study may provide guidance for a mechanism to explain the pathophysiological processes, particularly for both hyperacusis and Fragile X syndrome. For this, I thank the authors.

**Reviewer #2:**  The manuscript is well-written and scientifically sound. The conclusions are reasonable and not overreaching. There are very few minor typographical errors that should be corrected before publication. I am recommending this manuscript for publication. It adds to gaps in the scientific body of knowledge and will advance the FXS field forward.

6. PLOS authors have the option to publish the peer review history of their article (what does this mean? ). If published, this will include your full peer review and any attached files.

**Do you want your identity to be public for this peer review?** For information about this choice, including consent withdrawal, please see our Privacy Policy .

Reviewer #1: No

Reviewer #2: No

---

## [Author Response · Author response to Decision Letter 0]

5 Mar 2025

Response to Editorial Comments

Journal Requirement #1: Style Requirements

We have ensured the manuscript adheres to PLOS ONE’s style guidelines, referencing the provided templates (main body and title/authors/affiliations). Specific adjustments include reformatting the title page for readability (e.g., line break before email) and standardizing file names for submission.

Journal Requirement #2: Financial Disclosure

We have clarified the role of our funders by adding a Financial Disclosure statement after the Acknowledgments section:

"The present study was federally funded by the National Institutes of Health (NIH) Fragile X Center (U54HD082008 and U54HD104461) (PI: CE). The funders had no role in study design, data collection and analysis, decision to publish, or preparation of the manuscript."

This statement is also included in our updated cover letter.

Journal Requirement #3 and #4: Data Availability Statement

We have addressed the missing direct links in our Data Availability Statement and developed a comprehensive data sharing plan.

This statement is now included in the manuscript after the Acknowledgments:

Data Availability Statement

All deidentified data underlying the findings reported in this manuscript are publicly available in the following repositories:

- Raw EEG data from the study "Auditory EEG Biomarkers in Fragile X Syndrome: Clinical Relevance" (#1227) are stored in the National Database for Autism Research (NDAR), while source-localized tracings in EEGLAB SET format are deposited in Zenodo and can be accessed at 10.5281/zenodo.14967041.

- Custom code and scripts used to analyze data are available in a public GitHub repository at https://github.com/cincibrainlab/vhtp.

- Figures and analysis data tables are deposited in Figshare and can be accessed at 10.6084/m9.figshare.19435496.

These datasets and materials are provided in standard formats with accompanying documentation to facilitate reuse and replication. All data is deidentified and unrestricted, in accordance with PLOS ONE’s open data policy and our institutional ethics guidelines.

Journal Requirement #5: Supporting Information

We have removed supplementary figures and tables from the manuscript file and uploaded them separately as Supporting Information files (S1_Table.docx, S2_Table.docx, S1_Figure.pdf, S2_Figure.pdf, S3_Figure.pdf). Legends are now listed in a new section after the References:

"Supporting Information Legends: S1 Table. Assignment of cortical nodes... [etc.]"

Journal Requirement #6: Reference List

We reviewed the reference list for completeness and accuracy, confirming no cited papers have been retracted (checked via PubMed and Google Scholar as of March 04, 2025). Minor formatting adjustments were made.

Figure Files:

All figure files have been processed through the PACE tool to ensure compliance with PLOS ONE requirements and will be uploaded accordingly.

Response to Reviewers

Reviewer #1:

We thank Reviewer #1 for their positive assessment and for highlighting the study’s potential to guide understanding of pathophysiological processes in FXS and hyperacusis. While no specific changes were requested, we have emphasized the mechanistic implications in the Discussion (e.g., refined wording around gamma synchronization and frontotemporal connectivity) to align with your framing.

Reviewer #2:

We appreciate Reviewer #2’s endorsement of the manuscript as well-written, scientifically sound, and a valuable contribution to the FXS field. You noted “very few minor typographical errors” that should be corrected. We identified and fixed the following:

• Abstract: Added “the” before “temporal cortex” (p. 1).

• Introduction: Removed redundant “in” before “during” (p. 2).

• Materials and Methods: Added a space after “published” in citation [14] (p. 5).

• Results: Added “that” for clarity in STP prediction (p. 12).

• Discussion: Added “observed” before “significant” for flow (p. 18).

Additional Changes

To enhance clarity and readability, we made minor edits:

• Improved sentence flow in the Introduction (e.g., rephrased asynchronous gamma hypothesis, p. 2).

• Updated the manuscript to reflect Dr. Sweeney’s passing in the Acknowledgments, ensuring proper attribution.

We believe these revisions strengthen the manuscript and fully address all editorial and reviewer feedback. Thank you for your consideration. We look forward to hearing from you.

---

## [Editor Report · Decision Letter 1]

23 Apr 2025

Frontal Cortex Hyperactivation and Gamma Desynchrony in Fragile X Syndrome: Correlates of Auditory Hypersensitivity

PONE-D-24-20546R1

Dear Dr. Pedapati,

We’re pleased to inform you that your manuscript has been judged scientifically suitable for publication and will be formally accepted for publication once it meets all outstanding technical requirements.

Kind regards,

Abdolvahed Narmashiri

Academic Editor

PLOS ONE
---

## [Editor Report · Acceptance letter]

PONE-D-24-20546R1

PLOS ONE

Dear Dr. Pedapati,

I'm pleased to inform you that your manuscript has been deemed suitable for publication in PLOS ONE. Congratulations! Your manuscript is now being handed over to our production team.

Kind regards,

on behalf of

Dr. Abdolvahed Narmashiri

Academic Editor

PLOS ONE